# Genetic polymorphisms of muscular fitness in young healthy men

Tomas Venckunas[1]*, Hans Degens[1,2]

**1** Institute of Sport Science and Innovations, Lithuanian Sports University, Kaunas, Lithuania, **2** Department of Life Sciences, Musculoskeletal Science and Sports Medicine Research Centre, Institute of Sport, Manchester Metropolitan University, Manchester, United Kingdom

* tomas.venckunas@lsu.lt

**Data Availability Statement:** All relevant data are within the paper.

**Funding:** The author received no specific funding for this work.

## Abstract

The effects of genetic polymorphisms on muscle structure and function remain elusive. The present study tested for possible associations of 16 polymorphisms (across ten candidate genes) with fittness and skeletal muscle phenotypes in 17- to 37-year-old healthy Caucasian male endurance (n = 86), power/strength (n = 75) and team athletes (n = 60), and non-athletes (n = 218). Skeletal muscle function was measured with eight performance tests covering multiple aspects of muscular fitness. Along with body mass and height, the upper arm and limb girths, and maximal oxygen uptake were measured. Genotyping was conducted on DNA extracted from blood. Of the 16 polymorphisms studied, nine (spanning seven candidate genes and four gene families/signalling pathways) were independently associated with at least one skeletal muscle fitness measure (size or function, or both) measure and explained up to 4.1% of its variation. Five of the studied polymorphisms (activin- and adreno-receptors, as well as myosine light chain kinase 1) in a group of one to three combined with body height, age and/or group explained up to 20.4% of the variation of muscle function. *ACVR1B* (rs2854464) contributed 2.0–3.6% to explain up to 14.6% of limb proximal girths. The G allele (genotypes AG and GG) of the *ACVR1B* (rs2854464) polymorphism was significantly overrepresented among team (60.4%) and power (62.0%) athletes compared to controls (52.3%) and endurance athletes (39.2%), and G allele was also most consistently/frequently associated with muscle size and power. Overall, the investigated polymorphisms determined up to 4.1% of the variability of muscular fitness in healthy young humans.

## Introduction

As early as during adolescence, low skeletal muscle fitness poses a significant risk for chronic disease independent of cardiovascular fitness and overweight [1]. Although it is generally accepted that genetic variation accounts for >50% of the variability in muscle mass and function, and exercise performance [2–8], little is known about what specific genetic variants contribute to the ability to perform specific motor tasks, such as polymorphisms that predispose to high muscle strength and power. In addition, most studies performed to assess the relationship between polymorphisms and muscle function or exercise capacity lack precision in

**Competing interests:** The authors have declared that no competing interests exist.

phenotyping the participants, as detailed measurements with a battery of physiological and performance tests is time-consuming and challenging for both the subjects and the investigators. Yet, a study like this may help in "talent selection", where a good panel of polymorphisms may be used to direct people to sports they are "suited" best.

The ACE and ACTN3 genes are perhaps the most investigated genes and it has been observed that for instance the ACTN3 R-allele was associated with greater muscle power [9, 10]. It is equivocal whether there are such associations with ACE polymorphisms, where some studies do not find them [10] while the others did observe an association with muscle strength [11]. Perhaps the discrepancies between studies are related to differences in the parameters of muscle function that were studied and/or differences between populations, as Nazarov et al. [12] observed a higher prevalence of D-allele carriers in power athletes than endurance athletes and non-athletes.

The myostatin signalling pathway has emerged as one of the strongest candidates to explain the inter-individual variation in skeletal muscle phenotypes [13–15]. Indeed, muscle strength has been associated with polymorphisms in several genes involved in the myostatin signalling pathway [16–19]. For instance, the amount of lean body mass in older women has been found to associate with genetic variation in *ACVR2B* (rs2276541) [20] and the K153R polymorphism (rs1805086) of the myostatin (*MSTN*) gene has been associated with muscle strength in young men and women [21] and in aged women [22], and with muscle power in young men [23]. Another *MSTN* polymorphism (A55T) has been linked to muscle strength recovery after exercise-induced muscle damage in young men [24], and both K153R and A55T polymorphism have been associated with strength training-induced muscle hypertrophy [25]. As Mstn acts via its receptors *ACVR1B* and *ACVR2B*, it is of interest to know whether associations also exist between polymorphisms of the *ACVR1B* and *ACVR2B* genes with muscle size, strength and power.

Despite these reported associations between genotype and muscle mass and function, as illustrated above for the associations of muscle structure and function with genetic polymorphisms, so far studied SNPs have failed to explain even half of the estimated heritability for the trait [3, 26–28]. In fact, most investigated candidate genes have been shown to influence baseline muscle phenotypes as well as their response to training (i.e. muscle plasticity) only to a small extent at best [3, 10, 19, 29–31]. There thus still remains a long way to dissect the genetic factors that affect muscular fitness in humans.

To obtain a better understanding of the associations between genotype and muscle structure and function, in the current study we assessed the association of sixteen common genetic polymorphisms with a wide range of muscle fitness phenotypes in a large cohort of young healthy men varying in their training and fitness characteristics. Among the most investigated candidate genes *ACTN3* and *ACE*, we have also included genes of the myostatin signalling pathway (because of its primary relevance to muscularity), as well as those regulating myofibrillar contractile properties in smooth (*MYLK1*), skeletal (*MYLK2*) and cardiac (*MYLK3*) muscle, and putative modifiers of resting and exercise energy metabolism and muscle blood flow *via* differentiate sensitivity to catecholamine signalling (i.e. genes encoding adrenoreceptors *ADRA2A* and *ADRA2B*). We *hypothesized* that these less investigated candidate genes are at least as important as *ACTN3* and *ACE* for the muscular fitness.

## Materials and methods

### Subjects

The participants were 17- to 37-year-old Caucasian men from the Genetics and Epigenetics of Lithuanian Athletes from Kaunas (GELAK) cohort recruited as described previously [32, 33]. Potential participants were excluded from taking part in the study if they were known to suffer

**Table 1. Anthropometrics, muscular performance and cardiovascular fitness of the participants.**

| | C | E | P | T |
|---|---|---|---|---|
| Age (y) | 24.5 ± 4.3 (218) | 22.3 ± 3.6 (86)[a] | 22.7 ± 3.5 (75)[a] | 21.5 ± 3.1 (60)[a] |
| Height (m) | 1.80 ± 0.06 (216) | 1.79 ± 0.05 (86) | 1.81 ± 0.06 (75) | 1.87 ± 0.08 (58)[a,b,c] |
| Body mass (kg) | 77.4 ± 11.0 (218) | 70.5 ± 6.7 (85)[a] | 81.5 ± 11.9 (72)[a,b] | 80.8 ± 9.1 (59)[b] |
| BMI (kg·m$^{-2}$) | 23.8 ± 2.9 (215) | 21.9 ± 1.7 (84)[a] | 25.0 ± 2.8 (72)[a,b] | 23.1 ± 1.5 (58)[b,c] |
| Thigh girth (cm) | 54.5 ± 5.6 (217) | 53.0 ± 4.0 (86) | 57.5 ± 4.7 (75)[a,b] | 56.2± 2.8 (59)[b] |
| Upper arm girth (cm) | 30.9 ± 3.7 (217) | 28.0 ± 2.0 (86)[a] | 33.7 ± 3.7 (75)[a,b] | 31.2 ± 2.0 (59)[b,c] |
| Balance (no. of attempts) | 8.02 ± 4.50 (216) | 6.85 ± 7.14 (85) | 7.36 ± 4.73 (73) | 6.97 ± 3.34 (58) |
| Handgrip strength (kg) | 68.7 ± 10.1 (216) | 61.6 ± 8.7 (84)[a] | 70.6 ± 11.4 (73)[b] | 68.4 ± 8.8 (58)[b] |
| Knee extension PT (Nm) | 251 ± 47 (215) | 240 ± 37 (79)[a] | 301 ± 68 (72)[a,b] | 298 ± 46 (54)[a,b] |
| Knee flexion PT (Nm) | 135 ± 25 (215) | 134 ± 22 (79) | 152 ± 35 (72)[a,b] | 161 ± 25 (54)[a,b] |
| Vertical jump power (W·kg$^{-1}$) | 26.5 ± 2.0 (212) | 26.3 ± 2.0 (80) | 28.4 ± 2.3 (70)[a,b] | 28.1 ± 1.8 (55)[a,b] |
| Shuttle agility run (s) | 20.2 ± 1.3 (211) | 19.6 ± 0.9 (85)[a] | 19.7 ± 1.0 (73)[a] | 19.7 ± 1.4 (50) |
| 30 m run (s) | 4.56 ± 0.24 (214) | 4.43 ± 0.18 (84)[a] | 4.38 ± 0.19 (73)[a,b] | 4.31 ± 0.15 (58)[a,b,c] |
| Wingate (rev·30 s$^{-1}$) | 53.0 ± 5.8 (208) | 59.6 ± 5.0 (78)[a] | 56.6 ± 4.9 (72)[b] | 58.3 ± 4.6 (58)[a] |
| Wingate fatigue resistance (%) | 51.5 ± 7.9 (208) | 61.6 ± 8.2 (79)[a] | 53.8 ± 8.6 (72)[b] | 56.2 ± 7.5 (58)[a,b] |
| Pull-ups (no.) | 8.5 ± 4.8 (200) | 9.6 ± 3.8 (82) | 13.0 ± 5.0 (70)[a,b] | 9.0 ± 4.2 (56)[c] |
| VO$_2$max (ml·kg$^{-1}$·min$^{-1}$) | 50.8 ± 6.0 (186) | 64.1 ± 10.2 (74)[a] | 51.7 ± 5.8 (67)[b] | 54.6 ± 4.9 (48)[a,b] |

C: Control; E: Endurance athletes; P: Power athletes; T: Team athletes; Values are mean ± SD; between parentheses number of individuals; BMI: body mass index; PT: peak torque; VO$_2$max: maximal oxygen uptake.

[a]: different from control at $p \leq 0.024$

[b]: different from endurance athletes at $p < 0.05$

[c]: different from power athletes at $p < 0.05$.

from cardiovascular diseases or hypertension. The vast majority of the athletes were competitive, trained 3–14 times a week during the study, and were selected from the Registers of the Lithuanian Sports Federations. The study was approved by the Lithuanian National Committee for Bioethics, adhered to the guidelines of the declaration of Helsinki, and participants provided written informed consent before participation in the study. Athletes were divided into 3 groups as described previously [33]: endurance athletes (n = 86; distance runners, road cyclists, paddlers and skiers), power/strength athletes (n = 75; track and field sprinters, jumpers, throwers, combat sport athletes and body builders) and team athletes (n = 60; basketball, soccer and volleyball). Non-athletes (n = 218) trained for less than 2 h per week for the last 5 years. Characterization of the participants is provided in Table 1.

## Anthropometrics

Anthropometrics were determined before the performance tasks with the participants standing barefoot and wearing only shorts. Body height was measured using a metal stadiometer with the subject not touching any surface, and girths were measured with the flexible strap, both at 0.1 cm resolution. Dominant leg and dominant arm girths have been used for the analysis in this study. Body mass was measured after the overnight fast on a separate visit to the lab with electronic scales (TBF 300; Tanita, Tokyo, Japan) to the nearest 0.1 kg.

## Performance measurements

**Muscular fitness.** Subjects arrived at the lab at least 2 hours after the last meal. Before the tests, subjects warmed-up with 6–8 min of cycling on an electromechanically braked cycle

ergometer (Ergometrics–800S, Ergo Line, Medical Measurement Systems; Bitz, Germany) at ~70 rpm at the power in Watts equal to the body mass in kg, and then performed some light dynamic stretching exercises. The order of the tests was as follows: handgrip strength, Flamingo balance test, isokinetic dynamometry, pull-ups, countermovement jump, 10 x 5 m shuttle run, 30 m sprint run, and Wingate test.

## Handgrip strength

With the subject sitting with the hands on the table, handgrip strength at a comfortable grip position was measured with a hydraulic dynamometer (SH5001, Saehan Corporation, Korea) as the best of six attempts (three with each hand). The attempts were separated by ~1 min of rest, and to ensure a good grip of the dynamometer, a towel was used to dry/clean/absorb any sweat from the handle and the palms between the attempts.

*Balance*. The Flamingo balance test assesses one-leg standing ability on a 50 cm long, 5 cm high and 3 cm wide rigid beam [34]. In brief, the subject had to flex the free leg and grip the back of the foot with his hand of the same side, and after release of the support (and immediate start of the stopwatch by the observing researcher) try to remain on the beam in this position for 1 min. When balance is lost, (subject releases the suspended leg), the stopwatch is stopped and the test resumed without delay by the subject. The number of attempts needed to accumulate a total 60 s standing time is counted and considered inversely related to the static balance ability. Before the test, subjects were given one try with each leg to become familiar with the test and to decide which leg is dominant (more convenient to stand on), and that leg was used to start the test. Then the test was immediately performed on the other leg. For further analyses, the average number of attempts of each leg was used.

*Isokinetic dynamometry*. Knee extension and flexion torque of the dominant (the right one in most cases) leg was measured with the subject seated comfortably on a dynamometer (Biodex Pro3, USA) as described in more detail in Degens et al. [35]. The peak torques achieved for flexion and extension during three consecutive maximal effort full-range isokinetic flexion–extension contraction cycles at $30°·s^{-1}$ were used for further analysis.

*Pull-ups*. The number of full pull-ups (pronated grip) the subject is capable to perform in one set was measured, with the execution technique supervised (full extension of the arms at the end of eccentric phase, no leg swing to get make use of inertia, chin required to be raised above the bar) by a researcher who was also giving verbal encouragement to perform as many repetitions as possible with rests of no longer than 3 seconds between consecutive pull-ups. One attempt was allowed; subjects were not permitted to release the bar with either hand during the set. The bar was 3 cm in diameter, rigid and the width of the grasp was as preferred by the subject but corresponded to about the width of the shoulders.

As a measure of maximal leg extension *muscle power*, the participants performed three countermovement jumps (arms on the waist) separated by at least 1 min. The best jump was used for further analysis. Jump velocity at take-off ($v$ in $m·s^{-1}$) was calculated as:

$$v = a * t_f/2$$

where 'a' is the gravitational acceleration ($9.81 \ m·s^{-2}$) and '$t_f$' the flight time of the jump. The power was estimated as:

$$W = body \ mass * a \ x \ v$$

and expressed relative to a subject's body mass in kg.

To measure agility and *anaerobic power*, the subjects performed a 10 x 5 m shuttle run test according to the protocol of the Eurofit battery [34, 36], followed by a 30-m sprint run from a

standing start. Both running tests were conducted on concrete flooring, and the better of two attempts in the shuttle run and the best of the three attempts in 30-m sprint run were used for subsequent analysis. The time in the shuttle run was measured by an investigator with a stop-watch while it was measured electronically for the 30-m sprint.

The battery of the muscular fitness tests was concluded with a Wingate test to estimate anaerobic power and anaerobic endurance capacity [37]. In brief, the Wingate test was carried out on a mechanically-braked cycle ergometer (Monark 824E, Sweden) and preceded with fast unresisting acceleration to maximal pedaling cadence, after which the brake weight equaling 7.5% of body mass (to the nearest 0.1 kg) was applied at the signal of the subject, to initiate the 30-s all-out effort. Before and during the test, subjects were verbally encouraged to exert maxi-mally during the entire 30 seconds of the test. The number of full pedaling cycles per 30 s was then calculated, as was the power produced per each 5-s interval to derive a fatigue index, which is the percentage of power drop during the test.

### Aerobic capacity (cardiorespiratory fitness)

Aerobic capacity was measured with a $VO_2max$ test during a separate visit to the lab [35]. In brief, a ramp treadmill (H/P/Cosmos Sports & Medical GMBH, Germany) protocol of contin-uous incremental running until exhaustion was applied. After a warm-up of slow running for some minutes to familiarize with the treadmill, participants started the test by jogging at 7 $km \cdot h^{-1}$ for 3 min at an initial gradient of 1%, and then the speed of the treadmill belt was increasing by 0.1 $km \cdot h^{-1}$ each 6 seconds. The treadmill speed remained constant after it reached 20 $km \cdot h^{-1}$, and then the gradient of the treadmill was increased by 0.05% every 6 s from the initial 1%. Throughout the test, breath-by-breath gas analysis was performed using an Oxycon Mobile gas analyzer (Viasys, Germany), and heart rate (HR) was recorded with a HR meter (Polar 810s, Finland). During the later stages of the test, participants were verbally encouraged to keep running as long as they can. $VO_2max$ was calculated as highest average of 20 consecutive seconds.

### Genotyping

Venous blood was collected from an antecubital vein. DNA for genotyping was extracted from the venous blood samples using the NucleoSpin Blood kit (Macherey-Nagel, GmbH & Co. KG, Düren, Germany) according to the instructions by the manufacturer [37]. The polymor-phisms analysed are listed in Table 2.

*ACE* I/D (rs4341) genotypes were determined as described in Moran et al. [38]. *ACTN3* R577X (rs1815739), *MSTN* K153R (rs1805086), *ACVR1B* (rs2854464), *ACVR2B* (rs3792527 and rs7372545), *MYLK1* (C49T, rs2700352; H21P, rs28497577; and rs820336), *MYLK2* (rs6060965, rs4911532 and rs6119729), *MYLK3* (rs36471), *ADRA2A* (rs553668) and *ADRA2B* (rs4066772) genotypes were determined using PCR-RFLP described in detail in Moran et al. [38] and below.

### Copy number variation

Primer and hydrolysis probe assays (Integrated DNA Technologies Inc., USA) were designed to amplify a region lying in the introns of the genes of interest. The Basic Local Alignment Search Tool (BLAST, National Centre for Biotechnology Information) was used to ensure that the primers and probes were specific to the genes of interest and did not recognise any other sequences. All samples were run in triplicate and each plate contained a non-template control also run in triplicate. A reaction volume of 12.5 μL was used for all reactions, containing 6.25 μL of 2x Type-it CNV Probe PCR Master Mix (Qiagen Ltd, UK), 0.5 μL of 25x Ref assay,

**Table 2. Genetic polymorphisms analysed and genotype distribution (%) in C: Control; E: Endurance athletes; P: Power athletes; T: Team athletes.**

| | Genotype | C | E | P | T |
|---|---|---|---|---|---|
| ACE (rs4341) | II | 23.2 | 33.3 | 15.5 | 16.4 |
| | ID | 50.0 | 48.1 | 62.0 | 52.7 |
| | DD | 26.8 | 18.5 | 22.5 | 30.9 |
| MYLK1 (rs2700352) | AA | 0.5 | 1.2 | 1.4 | 0.0 |
| | AG | 32.0 | 32.1 | 26.8 | 21.8 |
| | GG | 67.5 | 66.7 | 71.8 | 78.2 |
| MYLK1 H21P (rs28497577) | GG | 82.5 | 85.2 | 90.1 | 81.8 |
| | GT | 17.0 | 13.6 | 9.9 | 18.2 |
| | TT | 0.5 | 1.2 | 0.0 | 0.0 |
| MYLK1 (rs820336) | CC | 8.2 | 6.2 | 9.9 | 7.3 |
| | TC | 42.3 | 42.0 | 35.2 | 36.4 |
| | TT | 49.5 | 51.9 | 54.9 | 56.4 |
| MYLK2 (rs6060965) | AA | 88.7 | 86.4 | 90.1 | 92.7 |
| | AG | 11.3 | 13.6 | 9.9 | 7.3 |
| | GG | 0.0 | 0.0 | 0.0 | 0.0 |
| MYLK2 (rs4911532) | CC | 29.4 | 35.8 | 39.4 | 30.9 |
| | CT | 54.1 | 40.7 | 45.1 | 50.9 |
| | TT | 16.5 | 23.5 | 15.5 | 18.2 |
| MYLK2 (rs6119729) | CC | 55.7 | 59.3 | 57.7 | 54.5 |
| | CT | 37.6 | 28.4 | 33.8 | 38.2 |
| | TT | 6.7 | 12.3 | 8.5 | 7.3 |
| MYLK3 (rs36471) | AA | 66.5 | 59.3 | 63.4 | 60.0 |
| | GA | 29.9 | 34.6 | 28.2 | 32.7 |
| | GG | 3.6 | 6.2 | 8.5 | 7.3 |
| MYLK3 V180L (rs28407821) | AA | 30.9 | 28.4 | 26.8 | 29.1 |
| | CA | 46.4 | 50.6 | 56.3 | 47.3 |
| | CC | 22.7 | 21.0 | 16.9 | 23.6 |
| MSTN K153R (rs1805086) | CC | 0.0 | 0.0 | 0.0 | 0.0 |
| | TC | 0.0 | 2.3 | 1.3 | 0.0 |
| | TT | 100.0 | 97.7 | 98.7 | 100.0 |
| ACVR1B (rs2854464) | AA | 47.7 | 60.8 | 38.0[b] | 39.6[a,b] |
| | GA | 39.9 | 30.4 | 43.7[b] | 32.1[a,b] |
| | GG | 12.4 | 8.9 | 18.3[b] | 28.[a,b] |
| ACVR2B (rs3792527) | CC | 36.3 | 27.5 | 34.8 | 21.8 |
| | TC | 50.0 | 47.5 | 39.1 | 49.1 |
| | TT | 13.7 | 25 | 26.1 | 29.1 |
| ACVR2B (rs7372545) | GG | 36.5 | 27.2 | 28.6 | 29.6 |
| | GT | 36.0 | 43.2 | 38.6 | 37.0 |
| | TT | 27.5 | 29.6 | 32.9 | 33.3 |
| ACTN3 R577X (rs1815739) | RR | 34.9 | 35.8 | 38.0 | 41.8 |
| | RX | 51.6 | 55.6 | 52.1 | 45.5 |
| | XX | 13.5 | 8.6 | 9.9 | 12.7 |
| ADRA2A (rs553668) | AA | 0.5 | 1.3 | 1.5 | 1.9 |
| | GA | 26.3 | 24.1 | 14.7 | 28.3 |
| | GG | 73.2 | 74.7 | 83.8 | 69.8 |
| ADRA2B (rs4066772) | II | 40.7 | 27.2 | 38.0 | 43.6 |
| | ID | 48.5 | 54.3 | 39.4 | 38.2 |

*(Continued)*

**Table 2.** (Continued)

| | Genotype | C | E | P | T |
|---|---|---|---|---|---|
| | DD | 10.8 | 18.5 | 22.5 | 18.2 |
| **Genotype score** | | 28.6 ± 2.4 (178) | 28.8 ± 2.4 (79) | 29.1 ± 2.4 (67) | 29.7 ± 2.3 (52) |

ACE: Angiotensin converting enzyme; MYLK1: myosin light chain kinase smooth muscle; MYLK2: MYLK skeletal muscle; MYLK3: MYLK cardiac muscle; MSTN: myostatin; ACVR: activin receptors; ACTN3: α-actinin-3; ADRA2: α-2-adrenergic receptor

[a]: different from control $P = 0.03$

[b]: different from endurance p≤0.037.

0.5 μl of 25x assay, 2.75 μl of nuclease-free water and 2 μl of DNA normalised to 5 ng/μL. All reactions were performed on the same LightCycler 480 (Roche Diagnostics Ltd, UK) system and included activation at 95˚C for 5 min followed by 40 cycles of alternative 30-s periods at 95˚C and 60˚C as recommended by the Type-it CNV probe protocol.

The genotype score was calculated as the sum of the scores for each genotype, where 3 was allocated to the homozygous genotype of a certain gene that according to the literature was expected to correlate with fitness positively, while a score of 1 reflected the homozygous genotype that was expected to correlate with fitness negatively. A score of two was given to the heterozygous genotype.

## Statistical analyses

The Shapiro–Wilk test showed that the data were normally distributed. An ANOVA with a Bonferroni-corrected post-hoc test was used to assess differences between groups. Hardy-Weinberg equilibrium was tested for all genotypes with a $\chi^2$ test. A Kruskal-Wallis test was performed to determine differences in the frequency distribution of polymorphisms between groups. Stepwise linear regression was performed with group, age, height and all polymorphisms as factors to assess which polymorphisms correlated with muscle fitness phenotypes Differences and correlations were considered significant at p<0.05. Phenotypic data is shown as mean ± SD.

## Results

### Participant characteristics and indices of fitness

Table 1 shows participant characteristics and fitness parameters. The BMI and limb girths were largest in power athletes and smallest in endurance athletes, while team sport athletes were the tallest (p<0.05). Team and power athletes had large knee extension and flexion torques, jump power and sprint running abilities than endurance athletes and controls (Table 1). Power athletes performed the largest number of pull-ups in 30 s. Anaerobic capacity, represented here as Wingate test average power and agility shuttle run time, was better in all three groups of athletes compared to controls; maximal oxygen uptake (VO$_2$max) and fatigue resistance in the Wingate anaerobic test were highest in endurance athletes.

### Genotype frequencies

Table 2 shows the distribution of genotype frequencies of the studied genetic polymorphisms in the three groups. All studied polymorphisms were in Hardy-Weinberg equilibrium in all groups. *ACVR1B* was the only polymorphism which differed between the groups with G allele overrepresented over A allele among team/power athletes compared to controls and

endurance athletes. Endurance athletes had a lower frequency of the GG genotype for the *AVCR1B* (rs2854464) gene than power (p = 0.037) and team (p = 0.004) athletes. In controls, the GG genotype frequency was also less than in team athletes (p = 0.03). There was no significant difference in genotype score between groups.

### Genotype effect on muscle size and exercise performance

Of the 16 polymorphisms tested, nine were associated with at least one of the muscular fitness and/or size phenotypes (Table 3).

The polymorphisms associated with muscular size/fitness were of seven genes (Table 3): *ACVR1B* plus *MYLK1* (rs2700352) explained 2.7% of thigh girth, and 3.3% of upper arm girth was explained by *ACVR1B*. For jumping power (W·kg$^{-1}$) the GG genotype associated with larger power (R$^2$ = 0.024; p = 0.002). Balance and pull-ups were predicted by the combination of *ADRA2A* and *ACTN3 R577X* for 2% and 1.8%, respectively. Knee extension torque was explained for 4.1% by *ACVR1B*, *MYLK1* (rs2700352), *MYLK1* (rs820336) and *MYLK3 (rs36471)*, and knee flexion torque for 1.7% by the *MYLK1* (rs2700352) polymorphism. The *MYLK1* H21P polymorphism, together with *ACVR2B* (rs379527) explained 3.1% to the variation in the shuttle agility run performance. The *ACVR2B* (rs379527) explained 4.0% of the variation of the Wingate test fatigue index.

Together with age, group and height as additional factors genotypes could explain up to 20.4% (knee extension torque) of the variation in muscular fitness (Table 4).

### Discussion

The current study on young Caucasian men, including athletes from various sports and healthy controls, found that nine of the 16 genetic polymorphisms tested and residing in regions coding for seven genes–myostatin, myostatin signaling receptors Activins 1B and 2B, myosin light chain kinases 1 and 3, alpha-actinin-3, and α-2-adrenergic receptor A–explained up to 4.1% of the variation of muscular fitness phenotypes either separately or when in combination. Activin receptor 1B was the only polymorphism that separated the groups, with the G allele overrepresented over the A allele among team/power athletes compared to controls and endurance athletes. The G allele of this gene was also most consistently associated with muscle size and power.

Multiple genetic polymorphisms have been identified to associate with the status of being a power athlete [30, 39]. However, the contribution of particular genetic markers for muscular fitness appears to be highly dependent on the specific characteristics of the population studied (age, ethnicity, athletic status etc.). For instance, five out of ten tested candidate SNPs were found to be associated with muscle power and sprint running performance in elite soccer players with their importance depending on the pubertal status of the athletes [40]. The polymorphisms associated with muscular fitness in our healthy young men irrespective of being a non-athlete, power, endurance or team athlete, therefore warrants testing for replication on other populations as aged people.

While myostatin (*MSTN*, *GDF-8*) is often included in power-oriented sport polygenic profiling [41], the K153R polymorphism with the strongest effect on muscle mass and strength is rather rare [42]. On its own, the K153R polymorphism explained up to 6.5% of the variance in some parameters of muscular fitness, even though only 3 subjects (two endurance and one power athlete) had the R allele in our cohort of 439 young men. The rarity of R allele in most of the world populations explains why most of the studies are underpowered to confirm phenotypic effects of this polymorphism [43–45]. However, polymorphisms of *ACVR1B* have been reported to modulate the response of muscle size and strength/power to rehabilitation of

**Table 3. Correlations between polymorphisms and fitness parameters.**

| Fitness parameter | ACE (rs4341) | MYLK1 (rs2700352) | MYLK1 H21P (rs28497577) | MYLK1 (rs820336) | MYLK2 (rs6060965) | MYLK2 (rs4911532) | MYLK2 (rs6119729) | MYLK3 (rs36471) | MYLK3 V180L (rs28407821) | MSTN K153R (rs1805086) | ACVR1B (rs2854464) | ACVR2B (rs3792527) | ACVR2B (rs7372545) | ACTN3 R577X (rs1815739) | ADRA2A (rs553668) | ADRA2B (rs4066772) |
|---|---|---|---|---|---|---|---|---|---|---|---|---|---|---|---|---|
| Thigh girth | R = -0.025; p = 0.611 | R = 0.168; p = 0.001<br>$R^2_{adj}$ = 0.027; p = 0.002 | R = -0.121; p = 0.015 | R = 0.077; p = 0.124 | R = -0.097; p = 0.052 | R = -0.020; p = 0.694 | R = 0.013; p = 0.788 | R = -0.032; p = 0.523 | R = 0.029; p = 0.558 | R = -0.177; $p<0.001$ | R = 0.127; p = 0.011 | R = -0.005; p = 0.922 | R = 0.012; p = 0.806 | R = -0.031; p = 0.539 | R = 0.107; p = 0.035 | R = -0.011; p = 0.830 |
| Upper arm girth | R = 0.067; p = 0.178 | R = 0.102; p = 0.041 | R = -0.081; p = 0.104 | R = 0.036; p = 0.471 | R = -0.030; p = 0.544 | R = -0.041; p = 0.412 | R = 0.015; p = 0.768 | R = 0.016; p = 0.741 | R = 0.051; p = 0.308 | R = -0.038; p = 0.444 | R = 0.175; $p<0.001$<br>$R^2_{adj}$ = 0.017; p = 0.007 | R = -0.070; p = 0.165 | R = -0.011; p = 0.828 | R = -0.004; p = 0.944 | R = 0.051; p = 0.316 | R = -0.014; p = 0.785 |
| Vertical jump power (W·kg$^{-1}$) | R = 0.002; p = 0.964 | R = 0.056; p = 0.265 | R = -0.115; p = 0.022 | R = 0.133; p = 0.008 | R = -0.098; p = 0.051 | R = 0.013; p = 0.793 | R = -0.002; p = 0.966 | R = -0.022; p = 0.659 | R = 0.020; p = 0.699 | R = -0.249; $p<0.001$ | R = 0.113; p = 0.026<br>$R^2_{adj}$ = 0.024; p = 0.002 | R = -0.002; p = 0.974 | R = -0.071; p = 0.165 | R = -0.035; p = 0.487 | R = 0.045; p = 0.375 | R = -0.033; p = 0.520 |
| Balance (no. of attempts) | R = 0.060; p = 0.231 | R = -0.024; p = 0.635 | R = 0.085; p = 0.090 | R = -0.095; p = 0.058 | R = 0.056; p = 0.262 | R = -0.031; p = 0.541 | R = -0.033; p = 0.505 | R = 0.060; p = 0.231 | R = -0.011; p = 0.827 | R = 0.255; $p<0.001$ | R = -0.011; p = 0.824 | R = -0.031; p = 0.542 | R = 0.019; p = 0.713 | R = -0.085; p = 0.091<br>$R^2_{adj}$ = 0.020; p = 0.010 | R = 0.043; p = 0.393<br>$R^2_{adj}$ = 0.011; p = 0.026 | R = -0.049; p = 0.324 |
| Handgrip strength | R = 0.061; p = 0.222 | R = -0.011; p = 0.821 | R = -0.011; p = 0.828 | R = 0.011; p = 0.833 | R = 0.018; p = 0.726 | R = -0.035; p = 0.481 | R = 0.076; p = 0.131 | R = -0.027; p = 0.588 | R = 0.006; p = 0.900 | R = -0.070; p = 0.165 | R = 0.092; p = 0.067 | R = -0.056; p = 0.269 | R = -0.068; p = 0.177 | R = 0.001; p = 0.988 | R = 0.038; p = 0.454 | R = -0.062; p = 0.220 |
| Knee extension peak torque | R = 0.026; p = 0.607 | R = 0.112; p = 0.027<br>$R^2_{adj}$ = 0.023; p = 0.005 | R = -0.043; p = 0.398 | R = 0.107; p = 0.035<br>$R^2_{adj}$ = 0.033; p = 0.002 | R = -0.078; p = 0.124 | R = -0.003; p = 0.956 | R = 0.055; p = 0.283 | R = 0.101; p = 0.046<br>$R^2_{adj}$ = 0.041; p = 0.001 | R = 0.049; p = 0.339 | R = -0.026; p = 0.614 | R = 0.114; p = 0.025<br>$R^2_{adj}$ = 0.013; p = 0.018 | R = -0.007; p = 0.899 | R = -0.025; p = 0.628 | R = -0.053; p = 0.296 | R = 0.033; p = 0.522 | R = -0.031; p = 0.542 |
| Knee flexion peak torque | R = 0.017; p = 0.738 | R = 0.152; p = 0.003 | R = -0.075; p = 0.140 | R = 0.064; p = 0.211 | R = -0.049; p = 0.339 | R = 0.027; p = 0.592 | R = 0.012; p = 0.818 | R = 0.041; p = 0.420 | R = 0.008; p = 0.873 | R = 0.032; p = 0.523 | R = 0.076; p = 0.139 | R = 0.074; p = 0.149 | R = 0.027; p = 0.602 | R = -0.024; p = 0.645 | R = 0.064; p = 0.214 | R = 0.016; p = 0.747 |
| Shuttle agility run (s) | R = -0.019; p = 0.713 | R = 0.069; p = 0.176 | R = 0.102; p = 0.044<br>$R^2_{adj}$ = 0.031; p = 0.001 | R = 0.038; p = 0.457 | R = 0.057; p = 0.265 | R = -0.077; p = 0.131 | R = 0.001; p = 0.982 | R = -0.027; p = 0.600 | R = 0.035; p = 0.486 | R = -0.047; p = 0.354 | R = 0.047; p = 0.361 | R = -0.149; p = 0.003<br>$R^2_{adj}$ = 0.020; p = 0.004 | R = -0.023; p = 0.660 | R = 0.042; p = 0.414 | R = 0.054; p = 0.296 | R = -0.090; p = 0.075 |
| 30 m run (s) | R = -0.026; p = 0.610 | R = -0.012; p = 0.809 | R = 0.045; p = 0.369 | R = -0.030; p = 0.546 | R = 0.045; p = 0.368 | R = -0.013; p = 0.789 | R = -0.003; p = 0.959 | R = 0.037; p = 0.466 | R = 0.024; p = 0.631 | R = -0.064; p = 0.205 | R = -0.089; p = 0.079 | R = -0.084; p = 0.097 | R = 0.017; p = 0.734 | R = 0.010; p = 0.844 | R = -0.046; p = 0.368 | R = -0.008; p = 0.868 |
| Wingate (rev·30 s$^{-1}$) | R = -0.066; p = 0.195 | R = -0.027; p = 0.599 | R = 0.051; p = 0.322 | R = 0.038; p = 0.451 | R = 0.054; p = 0.286 | R = -0.045; p = 0.378 | R = -0.011; p = 0.836 | R = -0.021; p = 0.682 | R = -0.063; p = 0.216 | R = 0.008; p = 0.875 | R = 0.090; p = 0.078 | R = 0.106; p = 0.039 | R = -0.056; p = 0.274 | R = -0.045; p = 0.381 | R = 0.046; p = 0.374 | R = -0.020; p = 0.690 |
| Wingate fatigue resistance | R = -0.066; p = 0.197 | R = 0.042; p = 0.410 | R = -0.083; p = 0.101 | R = 0.021; p = 0.677 | R = 0.017; p = 0.742 | R = 0.040; p = 0.429 | R = -0.026; p = 0.606 | R = 0.024; p = 0.632 | R = 0.038; p = 0.458 | R = -0.054; p = 0.287 | R = 0.027; p = 0.592 | R = 0.202; $p<0.001$<br>$R^2_{adj}$ = 0.040; $p<0.001$ | R = 0.077; p = 0.136 | R = -0.038; p = 0.459 | R = 0.006; p = 0.901 | R = 0.027; p = 0.596 |

(Continued)

**Table 3.** (Continued)

| | ACE (rs4341) | MYLK1 (rs27003352) | MYLK1 H21P (rs28497577) | MYLK1 (rs820336) | MYLK2 (rs6060965) | MYLK2 (rs4911532) | MYLK2 (rs6119729) | MYLK3 (rs36471) | MYLK3V180L (rs28407821) | MSTN K153R (rs1805086) | ACVR1B (rs2854464) | ACVR2B (rs3792527) | ACVR2B (rs7372545) | ACTN3 R577X (rs1815739) | ADRA2A (rs553668) | ADRA2B (rs4066772) |
|---|---|---|---|---|---|---|---|---|---|---|---|---|---|---|---|---|
| Pull-ups (no.) | R = 0.041; p = 0.423 | R = -0.036; p = 0.487 | R = -0.073; p = 0.154 | R = 0.009; p = 0.864 | R = 0.045; p = 0.378 | R = -0.052; p = 0.315 | R = 0.078; p = 0.130 | R = 0.049; p = 0.340 | R = -0.031; p = 0.539 | R = 0.002; p = 0.970 | R = -0.018; p = 0.729 | R = -0.002; p = 0.963 | R = -0.014; p = 0.791 | R = -0.086; p = 0.094 | R = -0.088; p = 0.092 | R = -0.034; p = 0.502 |
| | | | | | | | | | | | | | | $R^2_{adj}$ = 0.009; p = 0.045 | $R^2_{adj}$ = 0.018; p = 0.016 | |
| VO$_2$max (ml·kg$^{-1}$·min$^{-1}$) | R = -0.089; p = 0.097 | R = 0.004; p = 0.938 | R = -0.066; p = 0.216 | R = 0.054; p = 0.311 | R = -0.035; p = 0.513 | R = 0.055; p = 0.308 | R = -0.016; p = 0.760 | R = 0.016; p = 0.766 | R = -0.006; p = 0.904 | R = -0.216; p<0.001 | R = -0.069; p = 0.201 | R = 0.107; p = 0.047 | R = -0.005; p = 0.932 | R = 0.040; p = 0.460 | R = 0.057; p = 0.294 | R = 0.031; p = 0.562 |

If there is more than 1 row for a phenotype, the second row indicates which polymorphism is most explanatory in the darkest grey, with the contribution of additional polymorphisms explaining the phenotype to a lesser and lesser, but still significant, extent by lighter grey fields (showing increasing $R^2_{adj}$ and P values).

VO$_2$max, maximal oxygen uptake.

**Table 4. Adjusted determination coefficients ($R^2_{adj}$) of height, age, group and polymorphisms on the muscularity and fitness.**

| | Height | Age | Group | ACVR1B (rs2854464) | ACVR2B (rs379527) | MYLK1 (rs2700352) | MYLK1 H21P (rs28497577) | ADRA2B (rs4066772) |
|---|---|---|---|---|---|---|---|---|
| **Thigh girth** | 0.096*** | +0.006***→0.111 | +0.009***→0.105 | +0.020***→**0.131** | | | | |
| **Upper arm girth** | 0.057*** | +0.038***→0.095 | +0.017***→**0.146** | +0.036***→0.129 | | | | |
| **Handgrip strength** | 0.014*** | | | | | | | |
| **Vertical jump power (W·kg$^{-1}$)** | | | 0.080*** | +0.015***→**0.095** | | | | |
| **Isokinetic knee extension torque/ thigh girth** | 0.248*** | | +0.053***→**0.301** | | | | | |
| **Shuttle agility run (s)** | | | | | 0.018** | | +0.013**→0.031 | +0.008**→**0.039** |
| **Wingate fatigue resistance** | | +0.014***→**0.096** | 0.049*** | | +0.033**→0.082 | | | |
| **VO$_2$max (ml·min$^{-1}$·kg$^{-1}$)** | +0.009***→0.102 | 0.093*** | +0.009***→**0.120** | +0.009***→0.111 | | | | |
| **Knee extension peak torque** | +0.038***→0.188 | | 0.150*** | +0.008***→0.196 | | +0.008***→**0.204** | | |

*Note*. Only the significant determination coefficients

(*** @ $P < 0.001$

** @ $P < 0.01$) are presented. The coefficient of prime determinant of the phenotype is underscored, and the final estimation of the explained variance is highlighted **in bold**.

VO$_2$max, maximal oxygen uptake.

cardiac patients [46], and the A allele (rs2854464) has been reported to be associated with sprint/power performance, depending on the ethnicity of the athletes [47]. Although it has been reported that the genetic polymorphisms of the myostatin signalling pathway are more strongly associated with muscle strength than muscle mass [16, 48], in our cohort of young men the *ACVR1B* genotype was associated more with muscle size than strength (Table 3). While a positive association between *ACVR1B* A allele and muscle strength has been found in untrained men [18, 19], we found that the G allele was positively associated with muscle fitness phenotypes in our cohort of young individuals, and was also more prevalent among power and team sports athletes than in endurance athletes and non-athletes. This suggests that contradictory findings between the studies may occur also because of the different training status of subjects.

In another case-control study there was a trend towards lower frequency of *ACVR1B* A allele in Brazilian athletes, while Caucasian Power athletes had a higher frequency of the A allele compared to controls [47]. One may argue that these conflicting findings are due to different gene interactions in the two ethnicities, or just limitations of sample size. However, a

more likely explanation for the discrepancy is the dissimilar representation of the Power sport disciplines and training background between the two cohorts as in our Caucasian cohort we found that the A allele of the *ACVR1B* gene was underrepresented in Team sports athletes compared to Controls, and the A allele was also less frequent in both Team and Power compared to Endurance athletes. In this context, it is important to note that downregulated myostatin pathway activity is counterproductive for the endurance capacity [49, 50].

Caution should be exercised as to whether the inhibition of MSTN signaling benefits, as occurs in the *ACVRB1* G allele, muscle function since, for instance, the downregulation of activin 1 receptor (Alk4) expression not only decreased skeletal muscle mass to an even lower level in mdx mice (model of muscular dystrophy) but also induced marked muscle atrophy in normal animals [51]. In addition, lower myostatin signalling activity has recently been reported in patients with inflammatory myopathies as compared with healthy subjects, and there was no clear association between the activity of the myostatin signalling pathway and the functional state of the patients [52].

As with the ACE I/D polymorphism, studies on the effects of *ACTN3* R577X polymorphism on muscle characteristics have yielded so far inconsistent results [53], but the general impression is that the R-allele harbours benefits for sprinting performance and muscle power [54]. Also, the muscle damage response to intense exercise is not consistently associated with the *ACTN3* genotype [37, 55, 56]. Here we detected only a minor contribution of *ACTN3* R577X SNP to muscular fitness (adding not more than 1% of the explanation of the variance in two of the muscle fitness tests out of eight implemented) and any association to sprinting or jumping performance, which confirms the limited role of this genotype on strength and power in the healthy young population of European decent. This is in line with our previous observation that even though 2.4% of the variance in 40 m sprint run time has been explained *by* R577X polymorphism in Greek schoolboys, this was not the case in schoolgirls [38].

It has been suggested that polymorphisms of the skeletal muscle isoform of myosin light chain kinase (*MYLK2)* associates with exertional muscle damage [29] and may thus modulate the training response. It appears, however, that Clarkson et al. 2005 [29] did not study the skeletal muscle isoforms of MYLK and this may explain the discrepancy from the current study where we found no associations to muscular phenotypes with variants of *MYLK2*, while there were associations with genetic variants of *MYLK1* (smooth muscle isoform) and *MYLK3* (myocardial isoform). The *MYLK1* rs28497577 GT genotype has recently been shown to associate with slower recovery after knee injury in soccer players [57]. Intriguingly, polymorphisms of *MYLK1* might modulate the pro-inflammatory state of the bowel and nutrient absorption [58], which are important for the anabolic response in resistance training. It was surprising to see that polymorphism of cardiac myosin light chain kinase isoform (*MYLK3*) associates with muscle strength. This finding is new and needs confirmation by other studies. However, MYLK3 plays an important role in sarcomerogenesis in the heart [59] and its detection in skeletal muscle [60] suggests a similar role in skeletal muscle sarcomere assembly.

## Study limitations

One limitation is that we have pre-selected the set of candidate genes rather than screened for polymorphisms across the genome to pick up most of genetic markers associated with the muscular fitness of young healthy subjects. As in most other genotype-phenotype association studies, especially those of large cohorts of athletes, we have limited our study population to men and therefore direct extrapolation of the findings to women may not be applicable, as some other studies on functional SNPs on muscular fitness phenotypes have shown sexually dimorphic outcome [38, 61].

Study participants included into each of the Endurance, Power and Team sports category were from different sports and performance level. In addition, heterogeneity within the groups of athletes was high due to some of them being trained largely for their upper body while most of the athletes largely trained the leg or all body muscles. This disproportionally enhanced training of the upper body occurred particularly in the Power athletes where participants included not only sprinters but also body builders that clearly require different adaptations to succeed in their sport. However, they all trained their muscles with mostly high to maximal intensity short bursts of exercise, thus must develop overlapping adaptations which are somewhat contrasting to those of the endurance athletes.

Finally, while the study population used in the current study is smaller than that used in some association-seeking studies [62, 63]. In fact, the sample size was adequate to detect significant, and hence physiologically meaningful, associations, and one may query the physiological significance of any additional association if it can only be detected with investigating very large populations.

## Conclusions

Genetic polymorphisms in myostatin (*MSTNK153R*), the myostatin receptors Activins 1B (*ACVR1B*) and 2B (*ACVR2B*), myosin light chain kinases (MYLK) isoforms 1 and 3, and α-2-adrenergic receptor, and actinin (*ACTN3R577X*) were found to explain 1–4% of the variation in muscle fitness in young healthy male Caucasians, either separately or when combined. *ACVR1B* and *ACVR2B* polymorphisms associated independently to both muscle size and functional measures of muscular fitness, and the allele frequency distribution of *ACVR1B* rs2854464 differed between groups, with power and team athletes having higher frequency of the G allele. Variation in MYLK 1 and 3 genes contributes to some of the phenotypic variation of muscle size and power. Even though the contribution of these genetic polymorphisms to muscular fitness was quite large (up to 4.1%) given that only a limited number of gene polymorphisms were analysed, this also indicates that many more polymorphisms contribute to the 50% heredity of muscle mass and strength as has been recently shown by another group of researchers [64].

## Author Contributions

**Conceptualization:** Tomas Venckunas, Hans Degens.

**Data curation:** Tomas Venckunas.

**Formal analysis:** Tomas Venckunas, Hans Degens.

**Investigation:** Tomas Venckunas.

**Methodology:** Tomas Venckunas, Hans Degens.

**Resources:** Tomas Venckunas.

**Validation:** Tomas Venckunas.

**Writing – original draft:** Tomas Venckunas, Hans Degens.

**Writing – review & editing:** Tomas Venckunas, Hans Degens.

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
