## [Decision Letter · Decision Letter 0]

8 Aug 2022

PONE-D-22-13831Genetic polymorphisms of muscular fitness in young healthy menPLOS ONE

Dear Dr. Venckunas,

Thank you for submitting your manuscript to PLOS ONE. After careful consideration, we feel that it has merit but does not fully meet PLOS ONE’s publication criteria as it currently stands. Therefore, we invite you to submit a revised version of the manuscript that addresses the points raised during the review process.

We look forward to receiving your revised manuscript.

Kind regards,

Dalton Müller Pessôa Filho, Ph.D.

Academic Editor

PLOS ONE

Journal Requirements:

Reviewers' comments:

Reviewer's Responses to Questions

**Comments to the Author**

1. Is the manuscript technically sound, and do the data support the conclusions?

Reviewer #1: Yes

Reviewer #2: Yes

2. Has the statistical analysis been performed appropriately and rigorously? 

Reviewer #1: Yes

Reviewer #2: Yes

3. Have the authors made all data underlying the findings in their manuscript fully available?

Reviewer #1: Yes

Reviewer #2: Yes

4. Is the manuscript presented in an intelligible fashion and written in standard English?

Reviewer #1: Yes

Reviewer #2: Yes

5. Review Comments to the Author

Reviewer #1: The study intitled “Genetic polymorphisms of muscular 1 fitness in young healthy men” aimed to test for possible associations of 16 polymorphisms (across ten candidate genes) with fitness

and skeletal muscle phenotypes in 17- to 37-year-old healthy Caucasian male endurance, power/strength and team athletes, and non-athletes.

The paper is clearly written and deals with an important topic. The study of the association between gene polymorphisms related to team sports and performance tests is not totally original, specially ACTN3 and ACE, but the study design and the number of different genes allows some further clarifications. Used methods are appropriate and conclusions supported by presented results.

How was the number of athletes calculated? Was a statistical potency analysis conducted to demonstrate that the study size is adequate considering the expected SNPs distribution? The most recent papers are analyzing sample sizes much bigger than this present one.

Possible comments on ethical implications of an use of polimorphysms in athletic performance evaluation would enrich the discussion and help the reader in a more integrated perspective. What is the meaning of making these associations? How to use these kind of information to improve training effectiveness and avoid discrimination?

Reviewer #2: The authors in the introduction the authors emphasized the main genetic polymorphisms related to strength and muscle mass and the importance for their analysis. However, the other analyzed polymorphisms only mentioned its action. Should you explain how important it is for the analysis of these other polymorphisms in the introduction itself?

Paragraph between lines 45 and 49, the actors argue about the investigation of the two polymorphisms. But with the impact of this investigation? What kinds of benefits would the investigation of polymorphisms bring? It could be added in this paragraph.

Why did the authors opt for post hoc Bonferroni and not Tukey?

In the paragraph of lines 319 to 325, authors need the best explanation of the conflicting results.

In lines 352 and 353 "It was surprising to see that polymorphism of cardiac myosin light chain kinase isoform (MYLK3) associates with muscle strength. This finding is new and needs confirmation by other studies." What possible explanation for this? Authors could explain their result found.

In line 379 "limited number of gene polymorphisms were analysed, this also indicates that many more polymorphisms". What are these other polymorphism genes or the most important ones besides the polymorphism genes analyzed in the study?

6. PLOS authors have the option to publish the peer review history of their article (what does this mean?). If published, this will include your full peer review and any attached files.

Reviewer #1: **Yes: **Sandra L Amaral

Reviewer #2: No

---

## [Author Response · Author response to Decision Letter 0]

20 Aug 2022

Reviewer #1: The study intitled “Genetic polymorphisms of muscular 1 fitness in young healthy men” aimed to test for possible associations of 16 polymorphisms (across ten candidate genes) with fitness and skeletal muscle phenotypes in 17- to 37-year-old healthy Caucasian male endurance, power/strength and team athletes, and non-athletes. The paper is clearly written and deals with an important topic. The study of the association between gene polymorphisms related to team sports and performance tests is not totally original, specially ACTN3 and ACE, but the study design and the number of different genes allows some further clarifications. Used methods are appropriate and conclusions supported by presented results.

How was the number of athletes calculated? Was a statistical potency analysis conducted to demonstrate that the study size is adequate considering the expected SNPs distribution? The most recent papers are analyzing sample sizes much bigger than this present one.

Response: We agree that many studies have larger numbers of subjects recruited, but there are also still studies with similar numbers to assess gene polymorphism associations with muscle properties e.g. 

The Association of Multiple Gene Variants with Ageing Skeletal Muscle Phenotypes in Elderly Women. Khanal P, He L, Herbert AJ, Stebbings GK, Onambele-Pearson GL, Degens H, Morse CI, Thomis M, Williams AG. Genes (Basel). 2020 Dec 5;11(12):1459.

Prevalence and association of single nucleotide polymorphisms with sarcopenia in older women depends on definition. Khanal P, He L, Stebbings G, Onambele-Pearson GL, Degens H, Williams A, Thomis M, Morse CI. Sci Rep. 2020 Feb 19;10(1):2913.

Sarcopenia, Obesity, and Sarcopenic Obesity: Relationship with Skeletal Muscle Phenotypes and Single Nucleotide Polymorphisms. Khanal P, Williams AG, He L, Stebbings GK, Onambele-Pearson GL, Thomis M, Degens H, Morse CI.J Clin Med. 2021 Oct 25;10(21):4933.

Polygenic Models Partially Predict Muscle Size and Strength but Not Low Muscle Mass in Older Women. Khanal P, Morse CI, He L, Herbert AJ, Onambélé-Pearson GL, Degens H, Thomis M, Williams AG, Stebbings GK. Genes (Basel). 2022 May 30;13(6):982.

In addition, we measured a large set of characteristics of the participants that is time consuming and precludes the recruitment of thousands of people.

Nevertheless, even though the sample size was perhaps somewhat small for this sort of studies, it was adequate to detect significant, and hence physiologically meaningful, associations. In fact, one may query the physiological significance of any additional association if it can only be detected when studying large populations. We have added the following text to the ‘Limitations section’:

Finally, while the study population used in the current study is smaller than that used in some association-seeking studies, it is not of unusual sample size of many other similar studies (Khanal et al., 2020, 2022). In fact, the sample size was adequate to detect significant, and hence physiologically meaningful, associations, and one may query the physiological significance of any additional association if it can only be detected with investigating very large populations.

Possible comments on ethical implications of an use of polimorphysms in athletic performance evaluation would enrich the discussion and help the reader in a more integrated perspective. What is the meaning of making these associations? How to use these kind of information to improve training effectiveness and avoid discrimination?

Response: This is indeed an interesting question. Perhaps rather than helping to inform training strategies, it may help to identify whether someone is more likely to become a power or endurance athlete. In other words, it may help in a sort of positive "discrimination", better referred to as “talent selection”, where a good panel of polymorphisms may be used to direct people where they "suit" the best. It should be noted, however, that we did not primarily seek for a direct practical application of the results of this study, but rather to add to the discussion in sports genetics to predict athletic talents with a select number of SNPs. We have added the following text to the first paragraph of the Introduction:

Yet, such a study like this may help in “talent selection”, where a good panel of polymorphisms may be used to direct people to sports they are "suited" best.

Reviewer #2: In the introduction the authors emphasized the main genetic polymorphisms related to strength and muscle mass and the importance for their analysis. However, the other analyzed polymorphisms only mentioned its action. Should you explain how important it is for the analysis of these other polymorphisms in the introduction itself?

Response: We agree with the reviewer that there must be many more polymorphisms that contribute to muscular performance and other characteristics of exercise capacity. We had to limit, however, the number of polymorphisms to be studied for practical and financial reasons. We did, in fact, acknowledge that there are more SNPs that will contribute to muscle function and exercise capacity. We think that the explanation given there already signifies sufficiently that more work is needed with studying more polymorphisms such as those in the Piezo1 gene (Nakamichi et al., 2002; PMID: 35648809).

Paragraph between lines 45 and 49, the actors argue about the investigation of the two polymorphisms. But with the impact of this investigation? What kinds of benefits would the investigation of polymorphisms bring? It could be added in this paragraph.

Response: Thank you for this suggestion. We have added the following text to the first paragraph of the Introduction: 

Yet, such a study like this may help in “talent selection”, where a good panel of polymorphisms may be used to direct people to sports they are "suited" best.

Why did the authors opt for post hoc Bonferroni and not Tukey?

Response: We have chosen the Bonferroni post-hoc test, as it is more conservative, and hence if something was significant it is likely to be a false positive result.

In the paragraph of lines 319 to 325, authors need the best explanation of the conflicting results.

Response: We are not sure how we could improve the explanation of the apparent discrepancy between the observation in Caucasian and Brazilian athletes is that the study populations differed in the proportion of power athletes. This is what we wrote as a potential explanation:

However, a more likely explanation for the discrepancy is the dissimilar representation of the Power sport disciplines and training background between the two cohorts as in our Caucasian cohort we found that the A allele of the ACVR1B gene was underrepresented in Team sports athletes compared to Controls, and the A allele was also less frequent in both Team and Power compared to Endurance athletes

In lines 352 and 353 "It was surprising to see that polymorphism of cardiac myosin light chain kinase isoform (MYLK3) associates with muscle strength. This finding is new and needs confirmation by other studies." What possible explanation for this? Authors could explain their result found.

Response: Currently, little is known about impact of this gene on skeletal muscle structure and function, but it is suggested to be not only involved in sarcomerogenesis in the heart, but also in skeletal muscle (Liu et al., 2022). We therefore have added the following sentence to the Discussion for this gene:

However, MYLK3 plays an important role in sarcomerogenesis in the heart (Seguchi et al., 2007) and its detection in skeletal muscle (Liu et al., 2022) suggests a similar role in skeletal muscle sarcomere assembly.

In line 379 "limited number of gene polymorphisms were analysed, this also indicates that many more polymorphisms". What are these other polymorphism genes or the most important ones besides the polymorphism genes analyzed in the study?

Response: This could be many genes, and we think it is impossible to provide and extensive list here. It is at present unclear which ones are the most important genes, but one could be the mechanoreceptor Piezo1 (Nakamichi et al., 2002; PMID: 35648809).

---

## [Decision Letter · Decision Letter 1]

12 Sep 2022

Genetic polymorphisms of muscular fitness in young healthy men

PONE-D-22-13831R1

Dear Dr. Venckunas,

We’re pleased to inform you that your manuscript has been judged scientifically suitable for publication and will be formally accepted for publication once it meets all outstanding technical requirements.

Kind regards,

Dalton Müller Pessôa Filho, Ph.D.

Academic Editor

PLOS ONE

Additional Editor Comments (optional):

Reviewers' comments:

Reviewer's Responses to Questions

**Comments to the Author**

1. If the authors have adequately addressed your comments raised in a previous round of review and you feel that this manuscript is now acceptable for publication, you may indicate that here to bypass the “Comments to the Author” section, enter your conflict of interest statement in the “Confidential to Editor” section, and submit your "Accept" recommendation.

Reviewer #1: All comments have been addressed

Reviewer #2: All comments have been addressed

2. Is the manuscript technically sound, and do the data support the conclusions?

Reviewer #1: Yes

Reviewer #2: Yes

3. Has the statistical analysis been performed appropriately and rigorously? 

Reviewer #1: Yes

Reviewer #2: Yes

4. Have the authors made all data underlying the findings in their manuscript fully available?

Reviewer #1: Yes

Reviewer #2: Yes

5. Is the manuscript presented in an intelligible fashion and written in standard English?

Reviewer #1: Yes

Reviewer #2: Yes

6. Review Comments to the Author

Reviewer #1: The authors responded to all of this reviewer's questions and concerns. All changes are highlighted in the final version.

Reviewer #2: (No Response)

7. PLOS authors have the option to publish the peer review history of their article (what does this mean?). If published, this will include your full peer review and any attached files.

Reviewer #1: **Yes: **Sandra Lia Amaral

Reviewer #2: No

---

## [Editor Report · Acceptance letter]

19 Sep 2022

PONE-D-22-13831R1 

Genetic polymorphisms of muscular fitness in young healthy men 

Dear Dr. Venckunas:

I'm pleased to inform you that your manuscript has been deemed suitable for publication in PLOS ONE. Congratulations! Your manuscript is now with our production department. 

Kind regards, 

on behalf of

Prof. Dr. Dalton Müller Pessôa Filho 

Academic Editor

PLOS ONE
